# Exploring the Black Box of an mHealth Intervention (LIFE4YOUth): A Qualitative Process and Outcome Evaluation of End-User Engagement

**DOI:** 10.3390/ijerph192114022

**Published:** 2022-10-28

**Authors:** Anna Seiterö, Kristin Thomas, Marie Löf, Ulrika Müssener

**Affiliations:** 1Department of Health, Medicine and Caring Sciences, Linköping University, 581 83 Linköping, Sweden; 2Department of Biosciences and Nutrition, Karolinska Institute, 141 83 Huddinge, Sweden

**Keywords:** mHealth, mobile apps, behavior change, health interventions, user engagement, human–computer interaction, qualitative research, think-aloud

## Abstract

The effectiveness of mHealth interventions rely on whether the content successfully activate mechanisms necessary for behavior change. These mechanisms may be affected by end-users’ experience of the intervention content. The aim of this study was to explore how the content of a novel mHealth intervention (LIFE4YOUth) was understood, interpreted, and applied by high school students, and the consequences of engaging with the content. Qualitative content analysis was used inductively and deductively to analyze interview data (*n* = 16) based on think-aloud techniques with Swedish high school students aged 16–19 years. Theoretical constructs from social cognitive theory framed the deductive analysis. The analysis resulted in four categories which describe central activities of intervention engagement among end-users: defining, considering, centralizing, and personalizing. End-users engaged in these activities to different degrees as illustrated by four typologies: Literal, Vague, Rigid, and Creative engagement. Most informants knew about the risks and benefits of health behaviors, but engagement with intervention content generally increased informants’ awareness. In conclusion, this study provides in-depth knowledge on the cognitive process when engaging with mHealth content and suggests that deliberate and flexible engagement most likely deepens end-users’ understanding of why and how health behavior change can be managed.

## 1. Introduction

Health behaviors among adults are often influenced by habits formed during childhood and adolescence [1,2]. Behaviors such as physical activity, food habits, alcohol consumption, and smoking therefore not only influence quality of life among adolescents in the short term [3,4], but also very likely contribute to the prevalence of non-communicable diseases during adulthood [5]. Effective interventions are thus needed to promote healthy behaviors among adolescents who still form habits that will influence their health throughout life [6]. The term “mHealth” can be defined as the use of mobile technologies in public health and health service settings [7]. Interventions delivered via mobile phones, such as short text messages (SMS) and health applications, can reach many people and can be accessible to users at low cost, which has contributed to a rapid development of mobile phone based interventions over the past decade [8,9]. Several systematic reviews and meta-analyses have shown promising results of mHealth interventions on various health outcomes among youths [10,11,12], including health behaviors such as physical activity, food habits, alcohol consumption, and smoking [13,14,15,16,17].

Interventions must investigate the effect of mHealth tools on health outcomes, but also how these interventions work, for whom, and during which circumstances [18,19]. The term ”black box” usually refers to the mechanisms by which an intervention works, that is, how and why an intervention works [20]. For instance, an intervention may claim to increase physical activity in a population through increased self-efficacy. However, how intervention work and the impact of intervention mechanisms also depend on end-user engagement. The concept “end-user engagement” refers to the extent to which users make cognitive, emotional, and temporal investments to interact with mHealth content [21].

Process evaluations are recommended to achieve possible explanations of how or why intervention effects occur [19]. Systematic reviews show that process evaluations of behavior change interventions tend to focus on the delivery or implementation of interventions rather than examining the cognitive processes of user engagement, which is recognized as a limitation [22,23]. In-depth knowledge on how content is understood may be particularly important when interventions are delivered via mobile phones. Indeed, end-users of mHealth interventions typically use these interventions on their own, in contrast to face-to-face interventions which include on-going interaction and monitoring of the health care professional [24]. In addition, despite potential advantages of using qualitative methods to capture intervention-user interactions [25,26,27,28], systematic reviews show that most studies which have investigated users’ understanding, performance, and accuracy of intervention-related skills, such as the ability to recall information, are based on self-reported surveys [22,29].

The present study uses qualitative methods to explore cognitive processes of end-user engagement, which partly has been disregarded in previous research according to Rixon and colleagues [22]. Thus, this study seek in-depth understanding of how and why a novel mHealth intervention (LIFE4YOUth) works [30]. LIFE4YOUth addresses four health behaviors: physical activity, food habits, alcohol consumption, and smoking. The development of the intervention was guided by previous research in the area, health behavior theory, heuristic evaluations and usability tests, and focus group discussions with high school students [31,32,33]. This study is part of a larger project investigating the effects and implementation aspects of the LIFE4YOUth intervention. A randomized controlled trial is currently running to assess the effectiveness of LIFE4YOUth on health behavior change [30]. This study intends increase knowledge about the cognitive processes of how end-users’ engage with content, and about the consequences of the engagement. Thus, the aim of this study was to explore how high school students understand, interpret, and apply the content of an mHealth intervention (LIFE4YOUth) and to describe consequences on knowledge, outcome expectation, self-efficacy, perceived facilitators, and goals.

## 2. Methods

### 2.1. Design

A qualitative process and outcome evaluation [34] was performed by using think-aloud techniques [35] and semi-structured interviews with high school students aged 16–19 years. Qualitative content analysis [34] was used inductively to explore how end-users understood, interpreted, and applied intervention content, and deductively to describe the consequences of the engagement using theoretical constructs from social cognitive theory [36]. The consolidated criteria for reporting qualitative research [37] was used to ensure quality of reporting.

### 2.2. The LIFE4YOUth Intervention

The LIFE4YOUth intervention is delivered to end-users through weekly text messages. Each week, end-users will receive a text message prompting a brief weekly screening that includes all four health behaviors, followed by feedback on individual screening results in comparison with national guidelines. The feedback feature also provides access to intervention content (i.e., information and exercises) structured around two themes: (1) why to change health behaviors and (2) how to change health behaviors. The structure is similar in each of the four health behavior modules: physical activity, food habits, alcohol consumption, and smoking. End-users are able to freely navigate between modules. In addition, short text message services are available for each health behavior. The LIFE4YOUth intervention is described in detail elsewhere [30]. In this study, a wide selection of components in Life4YOUth was included to gain rich data on engagement characteristics. Intervention components were selected (Table 1) to represent a broad range and mixture of behavior change techniques and design characteristics (Appendix A).

### 2.3. Settings and Recruitment Process

The Swedish school system provides theoretical and vocational educational profiles. Theoretical studies traditionally prepare students for higher education, whereas vocational studies prepare students for professional practice and continued vocational training. Three urban high schools with a broad range of educational profiles and student characteristics (theoretical and vocational, females and males, and ages) located in the region of Östergötland, Sweden, were included. Eligible criteria were the ability to understand and be able to express oneself in Swedish, and not participating in the LIFE4YOUth effectiveness trial [30]. Informants were purposefully recruited in respect to age, gender, and educational profiles to achieve a varied sample set. Student recruitment was mainly performed by teachers who informed students about the study and administered contact between AS (Anna Seiterö) and interested students. In addition, AS visited two classes—one digitally and one physically—to recruit informants. All interested students were sent an e-mail with written information about the study. Those who afterwards confirmed their interest in participating were contacted by phone (by AS) to make an appointment for the interview. The recruitment took place between April and September 2021.

### 2.4. Informants and Data Collection

Thirty-one high school students notified researchers of their interest in participating. Of these, 12 students did not respond to either of the two e-mails that were sent. Two students responded that there was no time for participating. One student dropped out due to illness. Thus, individual interviews were performed with 16 high school students aged 16–19 (Table 2). All interviews took place online via video meetings in Zoom because of the COVID-19 pandemic. Interviews were conducted during (*n* = 3) or after (*n* = 13) school hours, depending on the agreement between students and teachers. No teachers were involved during the interviews. An interview guide developed for this study was used to guide data collection. Both the guide and the technical procedures were tested in a pilot interview with a representative 15-year student not participating in the study. No essential adjustments were made after the pilot interview.

Each interview was initiated informally and the interviewer’s (AS, female PhD student) role as a research coordinator and history as a former school nurse were described. There was no previous relationship between the informants and the interviewer. Before starting, all informants were encouraged to complete an exercise to be familiar with the think-aloud procedure [35]. The interview continued by giving informants first-time access to the intervention content by sharing screen with informants and asking informants to speak aloud what came to their minds while interacting with the content; for instance, while reading texts and completing exercises. Each informant was engaging with 6–8 intervention components (Table 1). The informants were continuously encouraged and reminded to think aloud. Probing questions such as “what made you respond in that way”, were used to elicit additional information. In addition, semi-structured questions such as “how did that exercise affect you?”, were asked after each informants’ interaction with the intervention content. Each interview lasted between 35 and 76 min (average 47 min). Notes to capture observations, such as non-verbal communication, were written during the interviews.

### 2.5. Ethical Approval

All informants received written and oral information about the study, including information that participation is voluntary, and that they are allowed to ask questions and leave the study at any time without giving reasons as to why. Informed consent was obtained from all informants orally during audio recording, in accordance with the ethical approval and guidelines. All procedures were approved by the Regional Ethical Committee (Dnr 2021-00136).

### 2.6. Data Analysis

Interviews were audio-recorded and transcribed verbatim by a professional firm. The analysis involved both inductive and deductive phases, starting with an inductive approach which explored how informants understood, interpreted, and applied the content. Next, the deductive phase used social cognitive theory [36] to structure data on engagement outcomes; knowledge, outcome expectation, self-efficacy, perceived facilitators, and goals.

First, in inductive analysis, all notes and transcripts were thoroughly read by AS, KT (Kristin Thomas), UM (Ulrika Müssener) (female PhDs). Essential information was noted in the margins (by AS) to inform the development of an inductive coding scheme [34] constructed in Microsoft Word. Coded text units were grouped and regrouped, which resulted in four activities characterizing informants’ engagement with the intervention content.

To further investigate contrasting patterns in the data, typologies were created based on the inductively generated categories and themes according to Patton [34]. Patton defines typologies as a series of patterns that are distilled into contrasting themes and ideal types. The characteristics of each typology was explored in terms of end-user engagement to offer illustrative end points of how end-users understood, interpreted, and applied the content.

In the final phase, a deductive coding scheme was developed based on theoretical concepts: knowledge, outcome expectation, self-efficacy, perceived facilitators, and goals [36]. Data from the semi-structured questions were coded, and text units were organized and assessed within each theoretical concept to describe how end-users’ engagement with the intervention contributed to outcomes. All analysis procedures have initially been performed by AS with contribution from UM and KT through discussions and conceptualization.

## 3. Results

These findings are based on interviews with 16 Swedish high school students aged 16–19 years during their first-time interaction with content in the LIFE4YOUth intervention. A majority of the informants had little experience of using mHealth tools but had experience of trying to engage in healthy lifestyle behaviors. Furthermore, most informants considered that they had been provided with health education within school recently (Table 2).

The results are presented in two sections. The first section describes the results regarding how informants understood, interpreted, and applied the content. The second section describes the results regarding consequences of engagement in terms of knowledge, outcome expectations, self-efficacy, perceived facilitators, and goals.

### 3.1. How the Intervention Was Understood, Interpreted, and Applied

Four categories were generated which describe how the informants processed and responded on information provided through the content in the LIFE4YOUth intervention: defining, considering, centralizing, and personalizing.

Defining of core concepts and acknowledging of previous knowledges and experiences was done more or less deliberately. For example, some informants deliberately decided how they should understand concepts, while others did not reflect at all on the meaning of concepts used in the intervention, and subsequently approached the intervention more hastily. Informants also differed as to whether they were considering content literally or whether they considered previous knowledge and experiences. Variations in considering mostly emerged when concrete examples were provided in the intervention. Some informants approached these templates rigidly, whereas others used them as inspiration or guidance (Table 3). The extent to which informants engaged in defining core concepts of the intervention (e.g., the meaning of vigorous physical activity) and the extent to which informants were considering previous knowledge and experiences contributed to how the intervention was understood and interpreted.

Furthermore, how informants understood and interpreted the intervention contributed to how it was applied (Figure 1). Data showed that in the process of applying content, informants differed in the extent to which they were centralizing themselves and personalizing the content. Centralizing refers to the point of departure that end-users take when attempting to apply intervention content. For example, some informants primarily approached content from their own perspective, whereas others approached content from the perspective of adolescents in general. Personalizing is about the extent to which informants adapted content to their specific circumstances. For example, some informants applied content literally, whereas others flexibly tailored the application to fit their own life situation (Table 3).

Deliberate engagement with defining and considering was recognized as key activities for understanding and interpretation of the content. A deliberate engagement can promote a flexible application, provided that personal circumstances are acknowledged through centralizing and personalizing. A deliberate and flexible engagement is overall understood as a cognitive investment which can result in beneficial consequences on psychological resources important for health behavior change (Figure 1).

Based on different patterns of understanding, interpretating, and applying content (illustrated by defining, considering, centralizing, and personalizing), four typologies were constructed: Literal, Vague, Rigid, and Creative (Figure 2). The typologies differed regarding to what extent engagement were deliberate and flexible. A high degree of deliberate and flexible engagement captures an overall high degree of reflection. All informants can be represented in more than one typology when interacting with different intervention content.

#### 3.1.1. Literal

The Literal typology represents a low degree of defining, considering, centralizing, and personalizing. This was illustrated as reading text without paying attention to how concepts were understood, but instead acknowledge explicit content, like examples. For instance, when reasoning about the number of portions of fruits eaten last week, some informants did not clearly define what a portion (of 100 g) means for fruits other than apples and bananas, which were used as examples. In addition, nothing other than fresh fruits were explicitly considered. Great emphasis was generally put on details of the content rather than what the content implies for themselves, which illustrates a low degree of centralizing. A consequence of applying content without deliberately defining and considering core concepts was that feedback derived from informant’s responses in some cases was assumed to imply something that was not truly accurate. This refers to low degree of personalizing.

#### 3.1.2. Vague

The Vague typology represents low degree of defining, a high degree of considering, a low degree of centralizing, and a high degree of personalizing. Thus, informants easily grasped the broad picture of what the content conveyed, illustrated by considering what they already knew about the subject and by spending brief period attempting to define concepts. Indeed, the engagement involved quick reading and sometimes missing textual details, such as reading out desired habits rather than undesired habits. Informants were flexible in how they approached content, such as not being restricted to specific examples or health behavior. The content was, however, applied at a general level because personal circumstances were not centralized, and goals and strategies lacked the precision to be personally useful. Some informants said that they would probably have spent more time on reflection if they perceived that their current habits were causing them any issues.

#### 3.1.3. Rigid

The Rigid typology represents a high degree of defining, a low degree of considering, a high degree of centralizing, and a low degree of personalizing. This typology was predominantly engaged in reflections and in reasoning about what concepts mean and how content should be understood. Content was approached rigidly, and informants considered the broader picture to a limited extent, for example reading between the lines. Narrow interpretation of the content sometimes affected informants’ assessment of credibility. For example, texts about vegetable consumption were, in some cases, interpreted as advocating for adolescents to become vegetarians, an idea with which some informants disagreed. In addition, some refrained from applying content that was perceived as not applicable or irrelevant to themselves based on the examples provided. This illustrates a high degree of centralizing, but a low degree of personalizing, because the possibility of writing free text was not utilized. Instead, some informants expressed a need for more diverse examples to capture all possible eventualities.

#### 3.1.4. Creative

The Creative typology represents a high degree of defining, a high degree of considering, a high degree of centralizing, and a high degree of personalizing. Informants processed information thoroughly and made deliberate decisions regarding how they defined and understood concepts. In addition, the content was processed as a whole, and knowledge gained from one section informed the understanding, interpretation, and application of other sections, which illustrates a high degree of considering. In addition, informants were consecutively reassessing whether something in their application (e.g., responses) should be changed as soon as new ideas appeared, which indicates high degree of centralizing. Informants’ understanding of how they could make use of the content therefore seemed to develop. In some cases, certain content was applied to a behavior (e.g., evening routines or procrastination behaviors) other than those addressed by the intervention. This illustrates how application of free text boxes was personalized to benefit personal needs.

### 3.2. Interaction Outcomes Based on Social Cognitive Theory

#### 3.2.1. Knowledge

Knowledge about health risks and benefits of different behaviors is considered a core determinant for health behavior change. Data showed that although most informants already knew about consequences of health behaviors and were familiar with the tips and strategies provided to them, they nevertheless perceived that factual information positively reminded them about something important. In fact, some informants perceived it to be an advantage that they recognized information, because it made them trust the intervention even more. The informational content elicited thoughts about habits and previous experiences, and some informants said that they became aware of perspectives or recommendations that they did not usually consider. Other informants emphasized that they did not need more knowledge. Instead, they stressed that they needed support to translate their knowledge into behavior.

“I thought that it would be normal, that I eat normal amounts. But it says that it’s…that I eat less than I should. I thought I had pretty good habits, but now when I think about it, I mean, how few vegetables I actually eat, maybe I should eat more than I do now”. (Informant 1: Female, 16 years).

#### 3.2.2. Outcome Expectation

The construct of outcome expectation is another central component of behavior change and involves weighing costs and benefits of different health habits. When interacting with the content, such as by reading texts and completing exercises, informants generally talked about their personal goals and desires in terms of a will to improve their wellbeing, feel more energized, and reduce stress. The interaction seemed to elicit thoughts about expectations linked to healthy habits, and many informants recognized that healthy behaviors are both a long term and short-term investment. Informants generally expressed that they achieved insights about varying perspectives, and many realized that there may be several reasons for them to adjust their habits.

“Because in the future…now I think very far ahead. Like, when I’m 50 years, have kids and…then you want to be able to do a lot of things with your kids, and not be that kind of mom who has to be in bed and have pain in the body”. (Informant 13: Female, 16 years).

#### 3.2.3. Self-Efficacy

Self-efficacy is described as a perception of being in control over one’s health habits. Some informants found the tips and strategies appealing because they were perceived as manageable regardless of their current health status. Informants often commented on their current habits when reading tips and strategies and recognized that they, to some extent, already had good habits. In addition, some justified engagement with health risk behaviors and said that it can be hard to admit such behaviors, especially if behaviors deviate from social norms. Informants who had recently made attempts to eat more fruit and vegetables also found feedback indicating too low of a consumption a bit annoying. Some content, however, increased informants’ awareness about what influences habits, such as when and why bad habits occur. Achieving such insights was perceived as comforting because that made their health risk behaviors more understandable. In addition, some informants realized that habits are nothing but behaviors, which are changeable. That also made them feel empowered.

“But it also gives me a better understanding, I haven’t thought much about these things before. And then, I mean…I go to school and am young, so it gives me much more control, and I can think about things that can happening and things like that”. (Informant 10: Female, 17 years).

#### 3.2.4. Perceived Facilitators

This construct involves social and structural facilitators and obstacles to the desired change. Data show that informants reflected on what changes might mean for them in relation to their friends, and some highlighted possibilities of being a role model. Others perceived that it would require determination to deviate from shared behaviors due to expectations and values of doing things together. Informants’ reflections about health behavior change also involved environmental aspects, such as whether biking is a possible transportation in their neighborhood. Several informants thought that the pandemic had increased their opportunity to be physically active during the daytime, but the boredom and laziness of being at home tended to be a hindrance, for example, of taking a walk during school breaks.

“…You reflect and realize “oh, maybe this [habit] wasn’t the best thing”. I mean, you’re thinking about what drives it [the behavior], and you think that…well, alright. Then you become aware about that you can also be affected by peer pressure. That it can actually happen to me, too”. (Informant 16: Male, 18 years).

#### 3.2.5. Goals

In comparison with outcome expectation, the goal construct involves concrete plans and strategies for realizing behavior change. Many informants who completed the goal-setting exercise expressed that it helped them specify what actions they need to do. Some had experience from similar goal setting strategies, while others had no experience from defining specific, measurable, accurate, relevant, and time-defined goals. Regardless of whether they accurately defined goals, several informants said that they lacked strategies for remembering to focus on their goal and therefore needed reminders. A few informants commented that they can create notifications themselves by using a mobile phone calendar, others thought they would require an mHealth tool.

“It is more like a planning for me, like…yeah, like when I should do it, for how long time I should do it, and whether it fits my everyday schedule”. (Informant 2: Female, 17 years).

## 4. Discussion

### 4.1. Result Discussion

#### 4.1.1. Main Findings

This study uses qualitative methods to evaluate the LIFE4YOUth intervention by exploring the black box in terms of how end-users understand, interpret, and apply intervention content, and consequences of the engagement. Qualitative evaluations of what happens when end-users engage with mHealth intervention content can contribute to a more comprehensive understanding about how specific interventions work, why, for whom, and under which circumstances.

Four activities (defining, considering, centralizing, and personalizing) were found to contribute to how informants understood, interpreted, and applied intervention content. In this study, defining is about figuring out what the content is about, i.e., acknowledging the map. Considering can be likened to an understanding of the map and a recognition of its possibilities. Centralizing addresses whether the map-holder finds out where they are positioned on the map and what the map implicates for themselves. Finally, personalizing is about making proper use of the map to navigate unfamiliar terrain.

These processes show similarities with Bloom and colleagues’ cognitive domains of learning: knowledge, comprehension, application, analysis, synthesis, and evaluation [40]. For example, defining concepts and considering previous experiences when approaching content requires an ability to recall and derive meaning from previous knowledge, and to consider this knowledge from various angles. Data showed that some informants used previous knowledge for reasoning about the credibility of the content and whether they agreed with the conveyed messages.

Despite the fact that this study did not focus on informants’ opinions regarding the content, or their interest in health behavior change, these findings suggest that end-users who consider themselves to be healthy or for other reasons have low interest in health behavior change, are likely to be found in the Vague typology. Health behavior theories emphasize that motivation is an essential determinant for health behavior change [41], and for engaging with mHealth interventions [42]. Indeed, personal relevance is considered decisive for continuing interaction with mHealth content [43], which might imply that characteristics found in the Vague typology can be connected to long-term non-engagement.

O’Brien and colleagues state that user engagement is a matter of quality rather than quantity with respect to cognitive, emotional, and behavioral investments regarding intervention interaction [21]. The typologies presented in this study can be seen as representations of the four types of engagement proposed by O’Brien. These stem from two crossed axes of engagement (negative to positive) and agency (low to high). For instance, the Creative typology described in this study clearly agrees with positive engagement and high agency. That is, high activity, motivated, interested, focused, and perceptions of being in control of the interaction. In addition, the Rigid typology shows similarities with O’Brian’s descriptions of negative engagement and high agency. That is, pausing, reflecting, being confused, and feeling challenged but still committed to persistence. Furthermore, the Vague typology shares characteristics with the negative engagement and low agency quadrant. That is, low activity due to, e.g., lacking motivation, but nevertheless completing tasks, which can result in passive learning [21].

#### 4.1.2. Implications for Practice

Typologies reported in this study are not grounded in characteristics of the informants, but on patterns regarding how the content was understood, interpreted, and applied. Content included in this investigation indeed has some differences regarding the character of the design. For instance, some content contains questions which end-users respond to through pre-determined choices and others by free text boxes. Additionally, only some content provides examples for the purpose of guiding end-users (Appendix A). It was not in the scope of this study to explore whether certain typologies are associated with certain characteristics of the content, or to certain BCTs. These findings nevertheless indicate that straightforward content, such as pre-defined statements only, would possibly limit engagement patterns seen in the Literal and Rigid typologies. However, access to less structured content, such as free text boxes, checklists, and guiding examples, seem to allow flexible application and perhaps facilitate engagement leading to useful end-user insights.

The findings of this study further illustrate two ways by which informants responded to content that could not be adapted. First, by being stimulated by the content, putting themselves in the center, and approaching the content as a tool for creatively exploring the circumstances which influence their behaviors (the Vague and Creative typologies). Secondly, by comparing themselves with conditions used as examples, and considering whether the content can or cannot be applicable to them (the Literal and Rigid typologies). Similarly, informants responded differently to content that could be adapted, such as tailored feedback. One typology in particular (the Literal typology) was characterized by poor critical appraisal of the validity of the feedback gained, e.g., not deliberately considering feedback in relation to their given responses. End-users’ difficulties in assessing tailored information, and mHealth tools’ restriction in tailoring content, highlight a significant difference between mHealth interventions and traditional face-to-face interventions. That is, the possibility of carefully adapting communication to meet the needs of specific persons. So far, most mHealth interventions do not include chatbots or other features built on artificial intelligence [24,44,45]. When these (hopefully) become more developed, they can help reduce hazards such as those identified in this study.

#### 4.1.3. Implications for Theory

The content investigated in this study was informed by social cognitive theory. According to Bandura, effective preventive interventions targeting adolescents should involve four components: information, self-management skills, self-efficacy, and social support. Self-management skills involve competencies that enable people to monitor their health, to know about influential circumstances, and to use goals in order to motivate and regulate behaviors [36]. Even though the LIFE4YOUth intervention are designed to affect at least all components but social support, these resources are assumed to develop over time. Yet, this study sought to increase knowledge about what impact the content could have on first-time end-users.

The analysis was framed by theoretical constructs from social cognitive theory: knowledge, outcome expectation, self-efficacy, perceived facilitators, and goals. The informants in this study generally perceived that they were already well-informed about the consequences of health behaviors. However, many informants realized something about their current habits (knowledge), attained a new perspective about why health behavior change might be desired (outcome expectation), and identified and thought about critical situations of relevance for their targeted goal (goals, perceived facilitators). That made some informants believe that they could better control their automatic behaviors in some situations (self-efficacy). This study does not respond to whether certain typologies or BCTs were associated with the quality of these outcomes. However, based on differences on how information was applied, the Creative typology is assumed to have a greater impact than the other typologies on the development of self-management skills and self-efficacy. Overall, the selected constructs are considered to provide a useful structure for describing aspects of how content affected end-users.

Indeed, qualitative studies allow for nuanced descriptions of mechanisms assumed to drive behavior change. The findings of this study can be useful when interpreting the results from other evaluations of the LIFE4YOUth intervention, such as quantitative mediator analysis and quantitative engagement analysis. Investigating the interaction processes echoes with the term receipt fidelity, which is often overlooked among intervention fidelity research [22,29]. The term “receipt” is contrasted with delivery and enactment by referring to an individual’s cognitive uptake of a delivered intervention as separate from behavioral responses [46]. In sum, the typologies reported in this study indicate that cognitive aspects should be emphasized in future research to better understand what end-users think, feel, and do when interacting with mHealth intervention content.

The typologies described in this study may be transferable to how other high school students understand, interpret, and apply content of the LIFE4YOUth intervention, and how the interaction possibly affects end-users’ knowledge, outcome expectations, self-efficacy, perceived facilitators, and goals.

Although think-aloud techniques are suggested to gain knowledge about engagement in mHealth interventions [47,48,49], so far, the techniques have mostly been used during formative evaluations of intervention development [50,51,52]. This study utilized the benefit of asking informants to verbalize their thoughts directly while interacting with intervention content to increase knowledge of how content can be understood, interpreted, and applied among Swedish high school students (i.e., the targeted population), and what consequences the engagement can result in. The think-aloud procedures contribute to methodological rigor [34] since all informants were interacting with the content in a structured way, and the role of the interviewer was less influential for achieving high quality data compared to in-depth interviews.

Informants were recruited from three different schools and heterogeneity were sought in respect to age, gender, and educational profiles. Data collection took place concurrently with data analysis, which is a strength in content analysis [53], and facilitates decisions of when to cease data collection. That is, cessation should occur when analytical ideas have been tested in interviews and still seem robust. Credibility was also strengthened by the involvement of three authors (AS, KT and UM) in the analysis [34], with competencies from different fields (nursing, public health, health psychology) and with different experiences of conducting qualitative research. Thus, with different lengths of experience and experience from different qualitative methodologies.

### 4.2. Limitations

A limitation of the study is that the results are based on an intervention-user interaction which occurred in research environment. That means that under real world circumstances, end-users may engage differently than they do when being observed. The credibility of the data was although considered by not informing the informants that AS was involved in the development of the intervention. The interviews took place online due to the COVID-19 pandemic, which also can have affected the informants more than if the interview had taken place during a physical meeting. That is, because screen sharing did not allow informants to actively click, scroll, and write, but only see the content on their screen and give instructions on where to click, what to write, etc. The intention with this study was to achieve in-depth understanding of the engagement process rather than assessing usability to inform intervention development. Hence, the consequence of not enabling informants to click and scroll themselves is considered less significant.

Furthermore, only some components of the LIFE4YOUth interventions were selected for this evaluation (Table 1), and effects on mechanisms essential for behavior change can also be affected by other components of the intervention, such as the text messages service. Because only one informant smoked cigarettes, this module was not included in order to protect the informant’s integrity. The structure of the content is however similar among the health behavior modules and the activities which formed the typologies are considered generic. Finally, because interviews were held with first-time end-users, the results do not essentially reflect how end-users would possibly engage with, or benefit from, the content over a longer period of time.

## 5. Conclusions

End-users understand, interpret, and apply LIFE4YOUth differently as illustrated by different degrees of defining, considering, centralizing, and personalizing when interacting with intervention content. These findings suggest that a deliberate and flexible engagement with mHealth interventions promotes good conditions for behavior change support. Differences in the degree to which end-users invest in cognitive processes may contribute to explaining why mHealth content is understood, interpreted, and applied differently among end-users. In addition, the findings suggest that end-users who deliberately and flexible acknowledged their specific circumstances, needs, and desires when interacting with intervention content can be empowered and achieve useful self-management skills. Furthermore, the study showed that think aloud techniques are a feasible method for collecting data on end-users thought processes when interacting with mHealth content. Finally, the findings propose empirically derived concepts which could inform future research and theory on end-user engagement within the mHealth research field.

## Figures and Tables

**Figure 1 ijerph-19-14022-f001:**
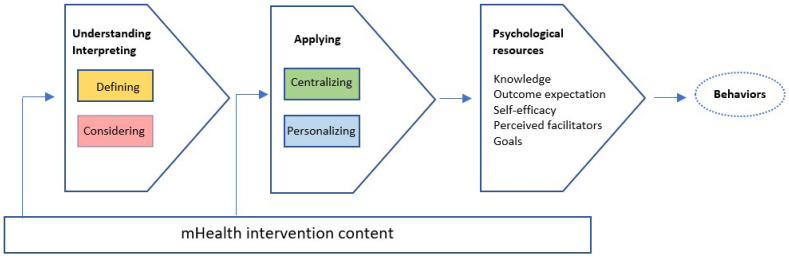
Model illustrating categories influencing end-users’ understanding, interpretation, and application of intervention content, which in turn may determine consequences of engagement.

**Figure 2 ijerph-19-14022-f002:**
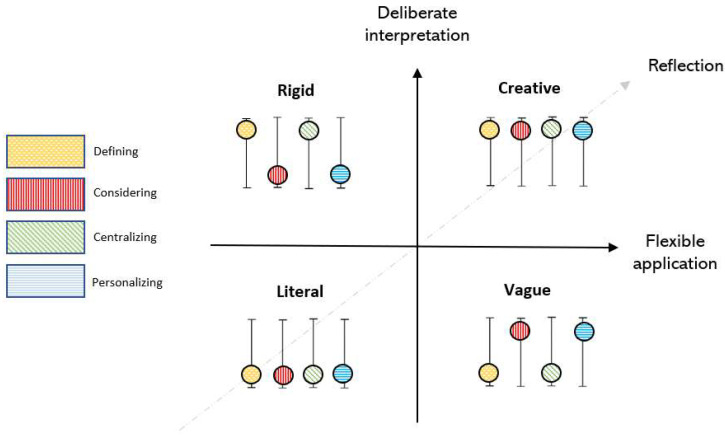
Typologies illustrating different patterns of deliberate and flexible engagement with an mHealth intervention (Life4YOUth).

**Table 1 ijerph-19-14022-t001:** Description of the content of LIFE4YOUth specified, per module.

Intervention Components	Design Characteristics	BCT ^1^	Physical Activity	Diet	Alcohol	Smoking
Weekly screening tool to prompt recording of health behaviors	Examples, predefined choices	Self-monitoring of behavior	x	x	x	x
Feedback on performance on behaviors	Colored feedback (green, yellow, red)	Feedback on behavior	x	x	x	x
Information to increase awareness of consequences of health behaviors	Text and pictures	Information about health consequences	x	x	x	x
Template to highlight consequences of behaviors	Predefined choices, examples, textual feedback	Information about social and environmental consequences	x	x		x
Prompting and identifying motives for health behavior change	Predefined choices	Incompatible beliefs	x	x	x	x
Practical tips to increase confidence in behavior change	Text	Instruction on how to perform a behavior	x	x	x	x
Instructions to define goals	Checklist, examples, free text boxes	Goal setting (behavior)	x	x		
Template to reflect on obstacles to behavior change	Examples, predefined choices, free text boxes	Problem solving Action planning	x	x	x	x

^1^ Based on the Behavior Change Techniques (BCT) Taxonomy [38].

**Table 2 ijerph-19-14022-t002:** Informant characteristics and health behavior modules addressed in interviews.

Variable	*n* (*n* = 16)
Sex	
Female	9
Male	7
Age, mean (range)	17.5 (16–19)
Educational profile	
Theoretical	12
Vocational	4
Satisfaction with life ^1^ (0–10)	
7–10	13
4–6	3
0–3	0
Prior experiences of using mHealth	
None	0
Little	14
Considerable	2
Prior experiences of attempts to improve	
Physical activity	13
Food habits	12
Smoking cigarettes	0
Alcohol consumption	1
None	1
Health education in school	
Recently	10
Not recently	6
LIFE4YOUth health behavior module engagement	
Physical activity	4
Food habits	9
Alcohol consumption	3
Smoking cigarettes	0

^1^ Cantril Ladder [39].

**Table 3 ijerph-19-14022-t003:** Illustration of how themes and categories are derived from data.

Theme	Categories	Example of High and Low Degree
Deliberate	Defining	“Well, let’s see, last week…yeah, I don’t know, maybe… I mean, I guess that we talk about alcohol here, right? Not just water and things like that.” (illustrates reflection about core concepts (standard units)) “Then it was…When thinking about soda, it was four, otherwise I only drink water.” (responded to a question in the weekly screening without realizing that the question referred to alcohol)
	Considering	“I usually don’t eat as much fruits as I did before, but I usually drink a lot of smoothies, though.” (How many 100 g portions (Equivalent to an average sized banana or one large apple) of fruit did you consume last week?) “I ate two bananas yesterday, so… I seldom eat apples.”
Flexible	Centralizing	“… I’m not sure about what’s considered a [Swedish fika]. Well, I am thinking about sweet stuff. But I use to buy sandwiches and so on in the café because I usually work out and must energize myself.” “I’m choosing one because…well, one can per week, that sounds normal.”
	Personalizing	“Yeah, which habit do I want to change? To sleep better, I mean, to get more sleep. It doesn’t have to be something related to exercise, right? It can be like sleep too?” “I…. right now, I’ve always eaten breakfast every day because I think it’s very important. […]. So, I maybe don’t have to put a goal on that, […] but maybe I should eat a bit more then, that could be a specific goal” (after reading about goal definition based on an example about having breakfast)

## Data Availability

Not applicable.

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
