# Peer review of "Exploring the Black Box of an mHealth Intervention (LIFE4YOUth): A Qualitative Process and Outcome Evaluation of End-User Engagement"

_ijerph, 2022, doi:10.3390/ijerph192114022_

Round 1

Reviewer 1 Report

I have read with a lot of interest the work entitled "Exploring the Black box of an mHealth Intervention 2 (LIFE4YOUth): a Qualitative Process and Outcome Evaluation 3 of End-user Engagement". I should recognize that the work provides findings about the mechanisms behind the success of mHealth interventions in solving youth medical matters. These findings confirm and build upon previous knowledge on this important topic. Despite the value of this contribution, several points should be solved before final publication:

1. Introduction: At the end of the Introduction, please provide an overview about what to expect from the paper so that the audience can find out what they will read and then can follow the paper.

2. Methods: Table 2 is difficult to follow. Please confirm enhancing its final layout. Subtitles like Prior experiences of using mHealth should be in the middle of the table and in bold font. Variable names should be aligned in the same way throughout the table. As well, although the study is partly observational, I do not find out why it is single-center (Three high schools from Östergötland, Sweden). Even in Östergötland, Sweden, more than three high schools could be easily involved. The authors could consider more than 31 students to have more than 16 participants to the experiment. The authors could explain why a few participants have been involved in this region. This will bring an added value to the generated results. One explanation can be the lack of interest of students in education and research as revealed at https://www.thelocal.se/20160913/here-are-the-best-and-worst-schools-in-sweden/.

3. Results: The Figure 1 should be moved so that it can be visible to the readers of the work. The explanation of Figure 1 and Table 2 can be expanded so that the discussion of the findings can be more relevant. Figure 1 and Table 2 are the main items that are used to motivate the findings and explain the main mechanisms behind the efficiency of mHealth applications. They deserve to be more discussed.

4. Conclusion: The Conclusion lacked future directions for this paper. As well, it lacked to reflect how the experiment is done and what are the results. Please feel free to expand the conclusion and add future directions for this research work based on the Discussion.

Based on these points, I invite the Editorial Board to accept this research paper after these major revisions are applied.

Reviewer 2 Report

The manuscript explores the black box of an mHealth intervention based on a qualitative process and outcome evaluation of end-user engagement. Although the paper is well-structured, a major revision is required. Please consider the following comments to improve the quality of the paper:

1.     I could not see the entire Fig. 1, please revise the format.

2.     L35-37, “Interventions delivered via mobile phones (mHealth)…”. This mHealth refers to mobile health not mobile phones, you should explain this in a professional way. Same requirement for short text message (SMS).

3.     L42, no need to use how in Italian. Same requirements for sub-titles.

4.     Shorten the title of Figure 2. It is too long to read.

5.     The novelty/originality of the paper should be more effectively established.

6.     Abstract should be improved by including the major findings of the work quantitatively, rather than qualitative description.

7.     You have provided several examples in the end of each sub-section of 3.2. I suggest restructuring them in the middle of the content. In this way, these examples can be supported, and the readability can be improved.

8.     I suggest only list limitations in Section 4.2 and rename it as limitations, the strengths can be separated and addressed in previous sections.

9.     Conclusion is too short. This makes the whole structure of the paper a bit weird.

10.  The legends in Fig 1 should be consistent with what shows in the figure.

11. L174 “by AS with contribution from UM and KT through…” I didn’t find their full names. I suggest that the authors either provide or add abbreviation list in the beginning to improve the readability. 

12.    Enlarge the text content in the supplementary document.

Round 2

Reviewer 1 Report

I thank the reviewers for having addressed all the comments regarding the manuscript entitled "Exploring the Black Box of an mHealth Intervention (LIFE4YOUth): a Qualitative Process and Outcome Evaluation of End-user Engagement". Based on this changes, I invite the Editorial Board to accept the paper for final publication.

Reviewer 2 Report

Thank you for your revision. I think the quality of the paper has been improved.